# Reprogramming biocatalytic futile cycles through computational engineering of stereochemical promiscuity to create an amine racemase

Sang-Woo Han [1,2], Youngho Jang[1], Jihyun Kook[1], Jeesu Jang[1] & Jong-Shik Shin [1] ✉

Repurposing the intrinsic properties of natural enzymes can offer a viable solution to current synthetic challenges through the development of novel biocatalytic processes. Although amino acid racemases are ubiquitous in living organisms, an amine racemase (AR) has not yet been discovered despite its synthetic potential for producing chiral amines. Here, we report the creation of an AR based on the serendipitous discovery that amine transaminases (ATAs) can perform stereoinversion of 2-aminobutane. Kinetic modeling revealed that the unexpected off-pathway activity results from stereochemically promiscuous futile cycles due to incomplete stereoselectivity for 2-aminobutane. This finding motivated us to engineer an $S$-selective ATA through in silico alanine scanning and empirical combinatorial mutations, creating an AR with broad substrate specificity. The resulting AR, carrying double point mutations, enables the racemization of both enantiomers of diverse chiral amines in the presence of a cognate ketone. This strategy may be generally applicable to a wide range of transaminases, paving the way for the development of new-to-nature racemases.

Chirality of organic compounds is a universal chemical code that underlies the biomolecular basis of living organisms[1]. Chiral symmetry breaking has been established in the construction of biosynthetic networks after the emergence of life by adopting a specific stereo-preference for biological building blocks, e.g. ʟ-amino acids for proteins and ᴅ-sugars for nucleic acids[2,3]. However, living organisms have also found a way to enrich metabolic and physiological diversity by allocating biochemical tasks to the non-preferred enantiomers that are readily available from the major chiral pool through stereoinversion[4]. For example, serine racemase supplies ᴅ-serine to mammalian brains as a signaling molecule[5,6]. This may explain why living organisms have evolved the enzymes specialized for racemizing certain chiral compounds, including α-amino acids and α-hydroxy acids[7].

The biological significance of chirality becomes crucial when ligand binding is controlled in a stereospecific way to target receptor proteins that are homochiral under ʟ-configuration[4,8]. In fact, the chirality of small-molecule pharmaceuticals plays a pivotal role in exhibiting desirable therapeutic efficacy, and therefore the chiral antipode of the effective enantiomer often causes serious side effects, as seen with Thalidomide[9]. Consequently, it is essential for the pharmaceutical industry to develop synthetic methods affording facile preparation of enantiopure chiral drugs[9].

Among the chiral building blocks present in active pharmaceutical ingredients, chiral amines constitute one of the essential stereogenic motifs and are found in approximately 40% of therapeutic drugs, including sertraline (Zoloft, depression), sitagliptin

[1]Department of Biotechnology, Yonsei University, 50 Yonsei-Ro, Seodaemun-Gu, Seoul 03722, South Korea. [2]Present address: Department of Biotechnology, Konkuk University, Chungju, South Korea. ✉e-mail: enzymo@yonsei.ac.kr

( Januvia, type 2 diabetes), rivastigmine (Exelon, Alzheimer's disease), tadalafil (Cialis, erectile dysfunction), and oseltamivir (Tamiflu, influenza)[10–12]. In addition, chiral amines are widely used for agrochemical synthesis[10]. As a result, extensive studies have been conducted on chemocatalytic enantioselective synthesis of chiral amines using organocatalysts and transition-metal catalysts via reductive amination of ketones or hydrogenation of imines and enamines[11–13]. It is also notable that traditional resolution methods through diastereomeric crystallization are still widely used despite a 50% yield limit[14]. However, growing social demands for sustainable and green processes have prioritized the development of biocatalytic methods[10].

Depending on the substrate type and the desired product outcome, the biocatalytic strategies available for chiral amine production are classified into kinetic resolution (KR), asymmetric synthesis, and deracemization, which exploit lipase, amine transaminase (ATA), monoamine oxidase, imine reductase, amine dehydrogenase, and reductive aminase[15]. Lipase is the only enzyme that does not act on the stereogenic C-N bond of the amine substrates and thus permits only KR[16]. However, despite the 50% yield penalty of KR, the lipase process is the most robust technology owing to its reliable scalability[16,17]. The yield limit problem of KR can be overcome by coupling in situ amine racemization with the enzymatic resolution, which has been the focus of extensive research efforts[16,18–21]. However, amines are rather recalcitrant to chemical racemization[22,23], and thus the resulting harsh reaction conditions are largely incompatible with biocatalysis[16,19,24,25]. As a result, the current lipase process relies on racemization and recycling of the unwanted amine leftover after KR[16].

An amine racemase (AR) would provide an ideal solution to address the biocompatibility issues associated with the chemical racemization. However, to date, such an enzyme, either natural or engineered, has not yet been reported[26]. Motivated by the potential utility of AR, we hypothesized that creating an AR might be possible by rerouting a native reaction pathway of ATAs, also known as ω-transaminases, as most racemases are evolutionarily and catalytically related to transaminases[27]. We serendipitously discovered that racemization of 2-aminobutane commenced at near equilibrium of ATA reactions, which piqued our interest in exploring the feasibility of creating an AR by engineering ATAs. In this study, we report on the computational active-site remodeling of an ATA to confer stereochemical promiscuity and demonstrate the racemization of both enantiomers of structurally diverse chiral amines using the engineered ATA.

## Results and discussion

### Theoretical feasibility of AR by engineering natural enzymes

To explore a plausible strategy for creating an AR, we compared two catalytically related enzymes: amino acid racemase (AAR) and ATA, both of which are pyridoxal 5′-phosphate (PLP)-dependent (Fig. 1a)[27]. AAR displays the desired reaction specificity, i.e. stereoinversion of Cα chirality, but does not accept amine substrates. On the other hand, ATA exhibits the matching substrate specificity but catalyzes a different reaction. Both enzymes share an initial reaction pathway leading to a carbanionic intermediate, and the ensuing steps bifurcate depending on the electron sink capacity of PLP (Fig. 1b)[28]. Unlike AAR, the protonated PLP of ATA enables negative charge delocalization and thus promotes protonation at C4′ of the carbanionic intermediate stabilized by a quinonoid structure (Supplementary Fig. 1)[28].

Based on mechanistic considerations, engineering of AAR to AR seems more feasible because it requires modification of only the substrate bias. However, the carboxylate group of amino acid substrates has been suggested to play a crucial role in the proton relay between catalytic bases of AAR[29]. This led us to discard the AAR strategy because an alternative proton transfer would be difficult to realize. On the other hand, engineering of ATA also poses challenges because it requires extensive mutations to shunt the canonical ATA pathway leading to the pyridoxamine 5′-phosphate form of enzyme (E-PMP), including adjusted electron sink capacity, complementary catalytic bases, and stereochemical promiscuity.

### Discovery of AR activity from native ATAs

We serendipitously discovered a clue to the engineering strategy when we found that the S-selective ATA from *Ochrobactrum anthropi* (ATA-OA)[30] converted S-2-aminobutane (S-**D2**) to the chiral antipode. We carried out transamination between equimolar S-α-methylbenzylamine (S-**D1**) and 2-butanone (**A2**), and monitored the generation of S-**D2** by chiral HPLC over a prolonged reaction time (Fig. 2a). Surprisingly, we found that S-**D2** produced by ATA-OA underwent a gradual reduction after 3 h, whereas R-**D2** was built up by the same amount of the S-**D2** decrement, with the total **D2** kept constant. This led to a continuous decrease in the enantiomeric excess (ee) of S-**D2** from >99% at 0.5 h to 62% at 24 h. We also examined the reverse reaction and observed the stereoinversion of S-**D2** again after reaching equilibrium (Fig. 2b), ending up with 58% $ee^S$ at 24 h. In contrast, stereoinversion of S-**D1** was not observed in both reactions (Supplementary Fig. 2).

To investigate whether the stereoinversion was independent of the cosubstrate, we tested seven additional amino donors (**D3**-**D9**) for

**Fig. 1 | Schematics of the engineering strategy and the reaction pathways.**
**a** Conceptual design of constructing AR using AAR and ATA as an engineering template. **b** Comparison of the whole reaction pathways of AAR and ATA as exemplified by a L-amino acid substrate of which Hα is colored red. The pyridine nitrogen is shown as an unprotonated form found in AAR.

reductive amination of **A2** (Fig. 2c). All the reactions resulted in progressive decreases in $ee^S$ of **D2** (Supplementary Fig. 3), leading to less than 30% $ee^S$ after 3 days. Even near-racemic outcomes of **D2** were attained with **D3, D5**, and **D7-D9**. It should be noted that the equilibrium position is dependent on the amino donor, resulting in different total **D2** concentrations.

The stereoinversion of $S$-**D2**, but not $S$-**D1**, suggests that the off-pathway activity results from incomplete stereoselectivity for **D2**. All ATAs are known to be strictly stereoselective for chiral amines and catalyze reductive amination of prochiral ketones with >99% $ee$, explained by a two-binding-site model consisting of a large (L) and a small (S) pocket (Fig. 2d)[31]. The S pocket has been proposed to accommodate up to an ethyl group, which is a crucial determinant for the stereospecific substrate recognition[31]. For example, productive binding of $R$-**D1** to $S$-selective ATAs is forbidden due to a steric clash of the phenyl group in the S pocket. Nevertheless, as we observed previously[32], the active-site model does not rule out the possibility that ATA-OA can marginally accept $R$-**D2** because both Cα-flanking substituents are no larger than an ethyl group. Consistent with this, $S$-3-methyl-2-butylamine did not show the stereoinversion due to an isopropyl substituent (Supplementary Fig. 4). However, the incomplete stereoselectivity for **D2** alone cannot explain the gradual changes in the chiral composition, because the stereoselectivity should remain constant regardless of the reaction progress.

### Kinetic dissection of ATA-OA

We conducted kinetic analysis of ATA-OA with each enantiomer of **D1** and **D2** to investigate the stereochemical properties (Fig. 2e). ATAs mediate two half reactions, namely oxidative deamination (OD) of an amino donor (D) and reductive amination (RA) of an acceptor (A), by shuttling between E-PLP and E-PMP[33]. Taking into account the reaction reversibility and substrate chirality, we examined four elemental reactions denoted by $I^S$, $I^R$, $II^S$, and $II^R$ where I and II represent **D1-A1** and **D2-A2** pairs, respectively. Note that going back and forth within each elemental reaction results in a futile cycle involving no overall change. A complete reaction can be expressed by coupling two elemental reactions. For example, $S$-**D1** + **A2** ↔ **A1** + $S$-**D2** is denoted by $I^S/II^S$ which represents $I^S$ coupled with the reverse of $II^S$.

Instead of using complicated nonlinear kinetics, we approximated that the OD and RA reactions follow second-order kinetics. For example, the reaction rate ($v$) for $I^S_{OD}$ is $k_{OD,I^S}$ [$S$-**D1**][E-PLP]. The strict $S$-selectivity for **D1** is manifested by the absence of generation of **A1** from $R$-**D1** and vice versa, leading to $k_{OD,I^R} ≈ 0$ and $k_{RA,I^R} ≈ 0$. It is worth noting that a $S$-**D1** impurity in the commercially available $R$-**D1** stock (determined to be 99.1% $ee^R$) initially resulted in generation of **A1** during OD of $R$-**D1**, but then **A1** plateaued at the same concentration as the $S$-**D1** impurity (Supplementary Fig. 5). In contrast, the incomplete stereoselectivity for **D2** enabled us to measure both $k_{OD,II^R}$ and $k_{RA,II^R}$, and we could determine the enantiomeric ratio (i.e., $E^S$ defined as $k^S/k^R$)[34] to be 20 for both OD and RA. The result of $E^S_{OD} = E^S_{RA}$ indicates that ATA-OA achieves the same stereocontrol over either direction of the reaction pathway.

The equilibrium constant ($K_{eq}$) of the elemental reactions can be determined by $K_{eq} = k_{OD}/k_{RA}$ because $v_{OD} = v_{RA}$ at equilibrium. The similar values of $K_{eq,II^S}$ and $K_{eq,II^R}$ in Fig. 2e are consistent with the notion that chemical equilibrium is irrespective of chirality. To verify this, we analyzed distribution changes in the enzyme intermediates upon exposure to each enantiomer of **D2** using spectral analysis (Fig. 2f). Incubation of the E-PLP form of ATA-OA with either $S$-**D2** or $R$-**D2** resulted in decreases in band 1, centered at 408 nm, and concomitant increases in band 2 corresponding to E-PMP ($λ_{max}$ = 339 nm)[35]. The spectral change induced by $S$-**D2** was faster than that by $R$-**D2**, consistent with $k_{OD,II^S} > k_{OD,II^R}$. Both samples reached equilibrium within 10 min as no further changes were detected. The identical spectra of ATA-OA equilibrated with $S$-**D2** and $R$-**D2** clearly indicate that population distributions of E-PLP and E-PMP are the same irrespective of the **D2** chirality, corroborating $K_{eq,II^S} = K_{OD,II^R}$.

### Computational simulations of the stereoinversion

To understand how ATA-OA achieves racemization of $S$-**D2** even with $E^S$ = 20 compared to $E ≈ 1$ of AARs, we carried out kinetic modeling of the reaction in Fig. 2a by considering $I^S$, $II^S$, and $II^R$. The simulation was initially carried out under suppressed futile cycle conditions by excluding $v_{I^S,RA}$, $v_{II^S,OD}$, and $v_{II^R,OD}$, leading to a constant $ee^S$ of **D2** throughout the reaction (Fig. 3a). This result had been thought to be an expected product outcome from stereochemically incomplete ATA reactions until we discovered the stereoinversion phenomenon.

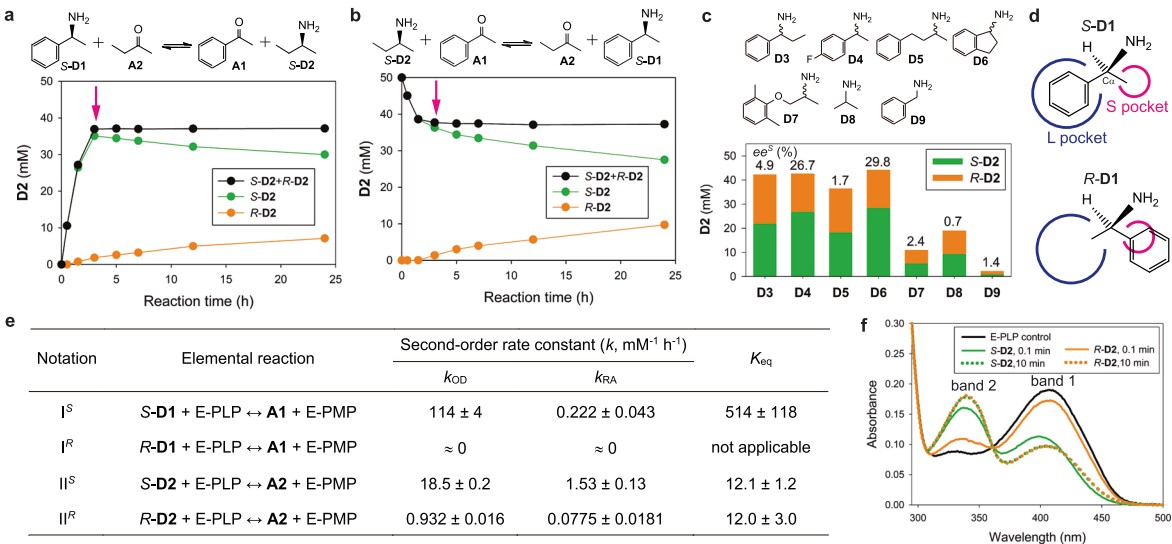

**Fig. 2 | Discovery and biochemical characterization of innate AR activity of ATA-OA for D2. a** Stereoinversion of the $S$-**D2** product from $S$-**D1** and **A2**. **b** Stereoinversion of the $S$-**D2** substrate in the reaction with **A1**. The red arrow in **a** and **b** indicates the time when the total **D2** starts to plateau. **c** Chiral composition of the **D2** product after 3-day reactions between **A2** and **D3-D9**. Numbers in the figure represent $ee^S$ (%) of **D2**. **d** Schematics of the active-site model of $S$-selective ATAs. **e** Stereochemical properties of ATA-OA examined by kinetic analysis for elemental reactions. **f** Spectral changes of the E-PLP form of ATA-OA induced $S$-**D2** and $R$-**D2**. Source data are provided as a Source Data file.

However, when the futile cycles were included in the simulation, reaction profiles resembled the experimental results and the predicted $ee^S$ was in reasonable agreement with the actual measurements (Fig. 3b). The simulation shows that $S$-**D2** begins to decrease at 3.8 h and complete racemization (i.e., $ee^S < 1\%$) occurs at 188 h (inset of Fig. 3b).

To better understand how the stereoinversion proceeds, we compare $v_{OD}$ and $v_{RA}$ of $I^S$, $II^S$, and $II^R$ (Fig. 3c). $I^S_{OD}$, $II^S_{RA}$, and $II^R_{RA}$ initially overwhelm the respective reverse reactions, and then $v_{OD}$ and $v_{RA}$ become closer to each other toward equilibrium. To trace how fast the three elemental reactions approach equilibrium, net reaction rates (i.e., $v = v_{OD} - v_{RA}$) are visualized in inset of Fig. 3c. As expected, $I^S$ approaches equilibrium (i.e., $v = 0$) most rapidly, seemingly because $k_{OD,I^S}$ is much higher than $k_{RA,II^S}$ and $k_{RA,II^R}$. In contrast, $II^S$ cannot approach equilibrium alone due to the chemical perturbation caused by the far-from-equilibrium state of $II^R$ which is occurring most slowly. As $II^R$ consumes **A2** under $v_{II^R,OD} < v_{II^R,RA}$, $II^S$ keeps adjusting $v_{II^S,OD}$ and $v_{II^S,RA}$ to compensate for the equilibrium perturbation. Eventually, $v_{II^S}$ changes the sign at 3.8 h and $II^S$ begins to convert $S$-**D2** into **A2** to fuel $v_{RA,II^R}$. Hence, stereoinversion via $II^S/II^R$, i.e. $S$-**D2** $\leftrightarrow R$-**D2**, begins at 3.8 h and proceeds until $S$-**D2** $= R$-**D2** to meet $k_{eq,I^S/II^R} = 1$.

### Direct racemization of D2 by coupling $II^S$ and $II^R$

As shown in Fig. 2b, the stereoinversion of $S$-**D2** in the presence of a non-cognate acceptor cannot lead to 100% yield of *rac*-**D2**. We posited that racemization of **D2** without the yield loss could be implemented by running only $II^S$ and $II^R$ in the presence of **A2**. Indeed, in agreement with the model prediction, both enantiomers of **D2** underwent desired racemization without concentration changes in total **D2** (Fig. 3d). Note that coupling $II^S$ and $II^R$ represents a futile cycle involving changes in chiral composition only. Intriguingly, the racemization of $S$-**D2** (top panel in Fig. 3d, $II^S/II^R$) was slower than that of $R$-**D2** (bottom panel, $II^R/II^S$) despite the $S$-preference of ATA-OA. This can be explained by the rate-determining step (RDS) of $II^S/II^R$ being slower than that of $II^R/II^S$. Numerical

(Fig. 2e). The $S$-to-$R$ inversion proceeds via $II^S_{OD}$ and $II^R_{RA}$, and thus is limited by $II^R_{RA}$ because $k_{OD,II^S} \gg k_{RA,II^R}$. However, the $R$-to-$S$ inversion is similarly limited by $II^R_{OD}$ and $II^S_{RA}$ as $k_{OD,II^R}$ and $k_{RA,II^S}$ are not very different. Therefore, $k_{RA,II^R}$ much lower than both $k_{OD,II^R}$ and $k_{RA,II^S}$ explains $v_{II^R/II^S} > v_{II^S/II^R}$.

To corroborate the RDS analysis, initial rates of $II^S/II^R$ and $II^R/II^S$ were measured at varying substrate concentrations (Supplementary Fig. 6). Indeed, $v_{II^S/II^R}$ was independent of $S$-**D2** but showed a positive correlation with **A2**, whereas $v_{II^R/II^S}$ was dependent on both $R$-**D2** and **A2**. Likewise, an increase in **A2** promotes racemization of both $S$-**D2** and $R$-**D2** with $II^R/II^S$ being faster than $II^S/II^R$ at the same **A2** level (Fig. 3e). This result led us to examine whether DMSO might augment the racemization reaction because RA of ketones by ATAs is known to be enhanced by DMSO[36]. Indeed, 15% (v/v) DMSO turned out to promote the racemization of both $S$-**D2** and $R$-**D2** (Supplementary Fig. 7).

To examine whether the racemization of **D2** is generally occurring among ATAs, we tested four additional enzymes, i.e. two $S$-selective (ATA-PD[31] and ATA-CV[37]) and two $R$-selective (ATA-AR[38] and ATA-AF[39]) ATAs cloned from *Paracoccus denitrificans*, *Chromobacterium violaceum*, *Arthrobacter* sp., and *Aspergillus fumigatus*, respectively. Indeed, all the ATAs exhibit AR activity for both $S$-**D2** and $R$-**D2** with the racemization of a preferred enantiomer being slower than that of the chiral antipode (Fig. 3f).

### In silico engineering of ATA-OA to accept $R$-D1

The AR activity discovered with natural ATAs results from incomplete stereoselectivity and thus can only be applied to a few amines with Cα-flanking substituents not larger than an ethyl group. Therefore, we set out to engineer ATA-OA using **D1** as a model substrate to create a generally applicable AR. We reasoned that racemization of **D1** by coupling $I^S$ and $I^R$ is thermodynamically feasible owing to $k_{eq,I^S/I^R} = 1$, but kinetically trapped due to $k_{I^R} \approx 0$. To reprogram the latent futile cycle, we opted to reduce the kinetic barrier of $I^R$. Numerical

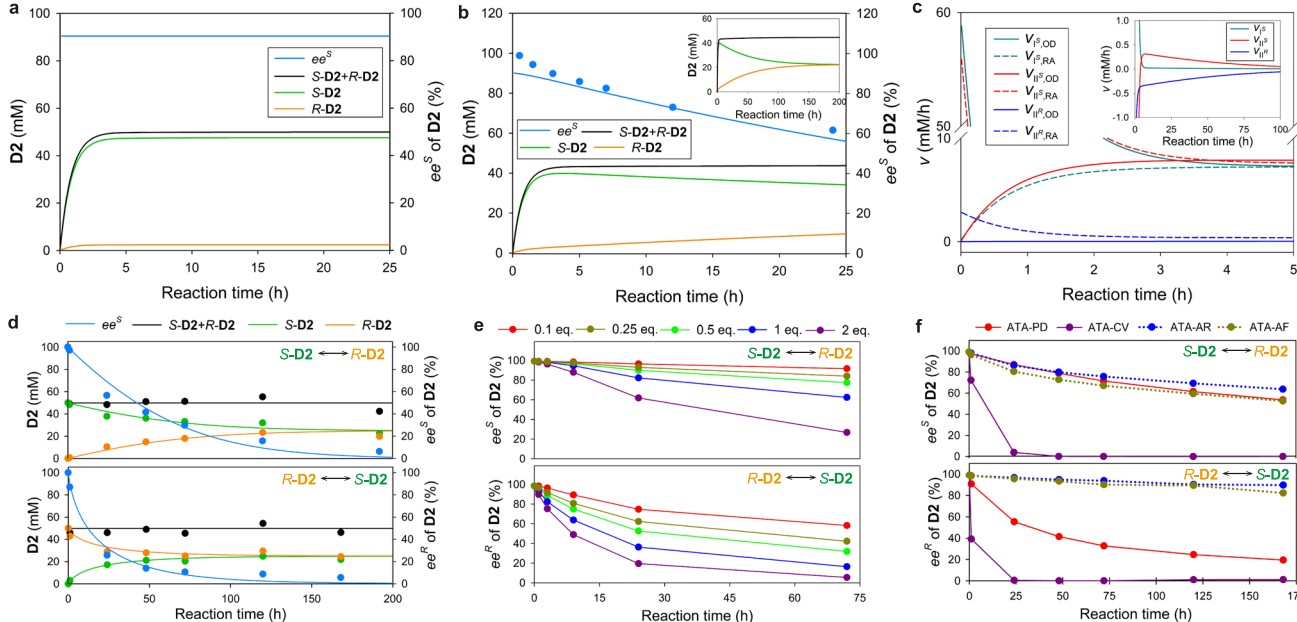

**Fig. 3 | Revealing kinetic basis of stereoinversion and demonstrating direct AR reactions by native ATAs. a** Kinetic modeling of the ATA-OA reaction in Fig. 2a by considering $v_{I^S,OD}$, $v_{II^S,RA}$, and $v_{II^R,RA}$ only. **b** Kinetic modeling of the same reaction by additionally including $v_{I^S,RA}$, $v_{II^S,OD}$, and $v_{II^R,OD}$. Closed circles represent experimental $ee^S$ adapted from Fig. 2a. The inset figure represents the reaction profiles up to 200 h. **c** Individual rate profiles of OD and RA for $I^S$, $II^S$, and $II^R$ for the simulation result of Fig. 3b. The inset figure represents the net rate profiles over an extended time. **d** Direct racemization of $S$-**D2** (top panel) and $R$-**D2** (bottom panel) by ATA-OA in the presence of **A2**. Solid lines represent simulation results. **e** Dependency of ATA-OA-catalyzed racemization of $S$-**D2** (top panel) and $R$-**D2** (bottom panel) on the **A2** concentration (0.1 – 2 molar equivalent to **D2**). **f** AR activity of various ATAs for $S$-**D2** (top panel) and $R$-**D2** (bottom panel) with **A2**. Source data are provided as a Source Data file.

simulations using a hypothetical ATA-OA with tunable $E^S$ ranging from 1 to 1000 show that both $I^S/I^R$ and $I^R/I^S$ proceed faster at lower $E^S$ and converge to an identical result at $E^S = 1$ (Supplementary Fig. 8).

In a previous study, we demonstrated computational engineering of ATA-OA to improve activities for ketones by exploiting mechanistic analysis of the nucleophilic attack trajectory (NAT)[31]. Here we applied the NAT analysis to identify an active-site residue whose mutation could allow productive binding of *R*-**D1** without hindering the native binding of *S*-**D1**. ATA-OA forms a homodimeric structure harboring two active sites on the subunit interface (inset of Fig. 4a). The residue R417, colored red, serves as a conformational switch that shifts between an outward structure to accept a bulky group and an inward one to recognize a carboxylate group[31]. The top docking pose of *S*-**D1**, ranked by binding energy, outlines the L and S pockets where the phenyl and methyl groups are accommodated, respectively (Fig. 4a). The asterisks of F86*, F323*, and T324* indicate that the residues are from the other subunit. The S pocket is located adjacent to PLP and is end-capped by S119, V154, and F323. The amino group of *S*-**D1** forms hydrogen bonds with the phenolic oxygen of PLP and π orbitals of Y151, seemingly increasing nucleophilicity of the substrate required for the nucleophilic attack on C4′ of PLP (Fig. 4b). The NAT length (i.e., Nα-C4′) of 3.1 Å is similar to 3.0 Å found with the reactive acceptor, e.g. benzaldehyde, in the previous study[31]. The angular orientation of NAT is represented by two characteristic angles (Fig. 4c), namely a dihedral angle ($\theta_{DH}$) and a Bürgi-Dunitz angle ($\theta_{BD}$), which are known to be optimal at 90° and 105°, respectively[40]. The docking pose of *S*-**D1** shows $\theta_{DH} = 78°$ and $\theta_{BD} = 91°$, close to the optimal orientations. Another crucial aspect to consider is the orientation of Hα relative to Nε of the catalytic lysine. After the nucleophilic attack, the ε-amino group of K287 released from the internal aldimine should be available to Hα for proton abstraction which is known to be a RDS[41]. This

explains why *R*-**D1** is completely non-reactive despite having a top docking pose similar to that of *S*-**D1** (Fig. 4d). Indeed, the NAT length, $\theta_{DH}$, and $\theta_{BD}$ of *R*-**D1** are 3.1 Å, 70°, and 79°, respectively. However, the spatial positioning assumes the relative orientation of Hα of *R*-**D1** opposite to that of *S*-**D1**. Therefore, the Hα of *R*-**D1** points away from Nε, making it inaccessible for proton abstraction. We inspected all the *R*-**D1** docking poses, but none of them showed the *S*-**D1**-like positioning of both Nα and Hα (Supplementary Fig. 9).

We also conducted docking simulations with *S*-**D2** and *R*-**D2** to investigate whether the NAT analysis could explain the incomplete stereoselectivity for **D2**. Among the docking poses of *S*-**D2**, the fourth one assumes desirable positioning of Nα and Hα (Fig. 4e), whereas the top three poses show limited accessibility of Hα to Nε. Regarding *R*-**D2**, the sixth pose allows productive binding where the ethyl and methyl substituents are accommodated in a reverse manner by the S and L pockets, respectively (Fig. 4f).

Taken together, we posit that productive binding of *R*-**D1** may occur if the S pocket is excavated large enough to accept a phenyl group. To explore this possibility, we performed in silico alanine substitution of the nine active-site residues shown in Fig. 4a, except for S119, A230, and R417, and then carried out docking simulations with *R*-**D1**. Note that the residues surrounding the methyl group of *S*-**D1** are Y20, L57, F86*, Y151, and T324*. Docking simulations predict that only the F86*A mutation allows for productive binding of *R*-**D1** (8th pose), considering spatial positioning of Nα and Hα (Fig. 4g). Truncation of the phenyl group from F86* upon the alanine substitution creates an enlarged S pocket with additional room indicated by a gray surface. This enables the reverse binding of *R*-**D1**, with the L and S pockets accommodating the methyl and phenyl groups of *R*-**D1**, respectively, similar to the binding of *R*-**D2** in the wild type. However, such a reverse binding of *R*-**D1** was not found with the other eight mutants, despite

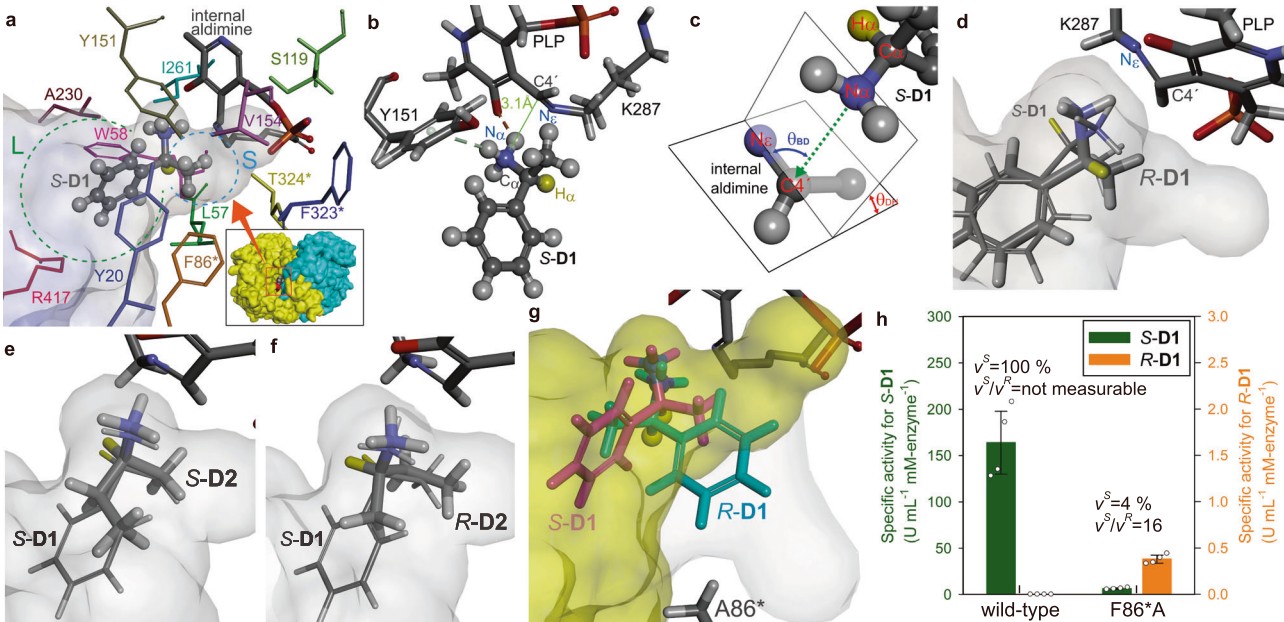

**Fig. 4 | Computational engineering of ATA-OA for stereochemical promiscuity toward D1. a** The top docking pose of *S*-**D1** in ATA-OA. Color use of the active-site residues is consistent with that of labels. The internal aldimine and *S*-**D1** are shown as thick-stick and ball-and-stick representations, respectively. The L and S pockets are enclosed by dotted circles colored green and cyan, respectively. The inset figure shows a surface representation of the homodimeric structure whose subunits are colored differently. The active-site entrance is marked by an orange square, with R417 highlighted in red. **b** A close-up view of the *S*-**D1** docking pose showing H-bonds (dotted lines) and the NAT length (green solid line). **c** Graphical depiction of NAT angular orientations. The green dotted arrow represents NAT. Docking

poses of (**d**) *R*-**D1**, (**e**) *S*-**D2**, and (**f**) *R*-**D2** in ATA-OA are shown as thick sticks, in comparison to that of *S*-**D1** (thin sticks). **g** Docking pose of *R*-**D1** (cyan) in the F86*A mutant of ATA-OA relative to that of *S*-**D1** in the wild type (magenta). Yellow and gray surfaces show the original active site and the expanded room created by F86*A, respectively. The Nα and Hα atoms of **D1** are shown as a ball representation and are colored blue and yellow, respectively. This coloring is consistent across **a** to **g**. **h** Activities of the F86*A mutant for *S*-**D1** and *R*-**D1** in comparison with those of the wild type. Note a hundred-fold difference in scale between the left and right y-axis. Source data are provided as a Source Data file. Data in **h** are mean values of quadruplicate experiments with error bars indicating the s. d. (*n* = 4).

thorough inspection of all the docking poses. The binding pose of S-**D1** in the F86*A mutant is almost identical to that in the wild type, suggesting that F86*A would not affect the native binding of S-**D1** (Supplementary Fig. 10).

To verify the model prediction, we carried out activity assays for R-**D1** with purified enzymes of all the nine mutants. Indeed, only the F86*A mutant exhibited a detectable activity for R-**D1** (Supplementary Table 1). The $v^S/v^R$ ratio was determined to be 16 (Fig. 4h). However, the F86*A mutation was found to eliminate most of the native activity, resulting in only 4% residual activity for S-**D1** compared to the wild type.

Considering the undetectable activity of the wild-type enzyme toward R-**D1**, the dramatic alteration in stereoselectivity induced by F86*A indicates that the bulky aromatic side chain of F86* serves as a structural gatekeeper against stereochemical promiscuity. To explore whether this stereocontrol strategy is shared among ATAs, we conducted a partial multiple sequence alignment of 14 S-selective ATAs whose X-ray structures have been determined (Supplementary Table 2). Indeed, the F86* residue of ATA-OA is conserved and represented by aromatic amino acids, specifically eleven Phe, two Tyr, and one Trp.

To verify whether the reduction in size of the gatekeeper residue could lead to deteriorated stereocontrol in other ATAs, we measured the stereoselectivity of an F85*A mutant of ATA-PD for **D1** in comparison with the parental enzyme (Supplementary Fig. 11). Unlike ATA-OA, ATA-PD displayed a detectable basal activity for R-**D1** and the resulting $v^S/v^R$ ratio of the wild-type enzyme was determined to be $4100 \pm 400$. As observed with ATA-OA/F86*A, the F85*A mutation introduced to ATA-PD led to a drastic loss of stereocontrol for **D1**, representing a 15-fold reduction in $v^S/v^R$. Notably, the ATA-PD/F85*A mutant showed significantly reduced activity for S-**D1**, at only 9% residual activity, similar to ATA-OA/F86*A.

In addition to ATA-OA and ATA-PD in this study, it was reported elsewhere that ATA-CV underwent a remarkable reduction in stereoselectivity for **D1** due to F88*A mutation, resulting in a change in the E value from 150 to 7[42]. In line with these results, Ao et al. recently reported that a point mutation to leucine at the same position resulted in ATA from *Ruegeria* sp. TM1040 exhibiting substantial activity for R-**D1**[43].

## Combinatorial mutations to restore native activity

The drastic loss of the native activity of ATA-OA caused by F86*A, despite the intended stereochemical promiscuity, prompted us to carry out a second round of enzyme engineering to restore the activity. The active site of ATA-OA is mostly constructed by a single subunit and is completed by recruiting a loop region of another subunit from F82* to R89* (Fig. 5a). Notably, F86* directly participates in the active site by filling in the breach between L57 and Y20 (Fig. 5b). Therefore, we suspected that F86*A might be detrimental to the structural stability of the active site. To verify this presumption, we tested F86*L for **D1**, anticipating a milder structural perturbation and, consequently, a lower loss of activity compared to what was observed with F86*A (Supplementary Fig. 12). Indeed, the F86*L mutant showed only a 20% reduction in native activity for S-**D1**. It is worth noting that F86*L allows for substantial activity for R-**D1**, resulting in $v^S/v^R = 91$, reminiscent of the Y87*L mutant of ATA from *Ruegeria* sp. TM1040[43].

In a previous study, four active-site residues, i.e. L57, W58, V154, and I261, were identified as activity-improving spots upon alanine substitution, and their combinatorial mutations were found to be additive[44]. Thus, we expected that incorporating a second alanine substitution might offset the F86*A-induced structural perturbation. Indeed, the additional mutation led to the desired activity recovery, and W58A and V154A elicited remarkable activity improvements for S-**D1** by 48- and 14-folds, respectively, compared to the F86*A mutant (Fig. 5c). We chose the F86*A/V154A mutant for further study because

of the undesirably high $v^S/v^R$ of the W58A/F86*A mutant. Notably, stereochemical promiscuity of the F86*A mutant is reinforced by the addition of V154A.

Kinetic study of the F86*A/V154A mutant, named AR-OA, reveals similar $E^S$ for $I_{OD}$ and $I_{RA}$ (Fig. 5d). Likewise, $E^S$ for $II_{OD}$ and $II_{RA}$ are similar, as observed with ATA-OA. All the four $E^S$ values are not very different from unity, reminiscent of AARs. It is also noteworthy that the $K_{eq}$ values for the chirally opposite reactions are in reasonable agreement. The similar $K_{eq}$ values for $I^S$ and $I^R$ were verified by the spectral population analysis of AR-OA equilibrated with S-**D1** and R-**D1** (Supplementary Fig. 13).

As expected, AR-OA is capable of catalyzing racemization of S-**D1** which is not observed at all with ATA-OA (Fig. 5e). Supplementation of the reaction mixture with 15% DMSO promoted the racemization of S-**D1** (Fig. 5f), as observed with that of **D2** by ATA-OA (Supplementary Fig. 7). AR-OA affords racemization of S-**D2** even faster than ATA-OA does (Supplementary Fig. 14), seemingly owing to the 4.5-fold increase in $k_{RA,II^R}$.

As shown in Fig. 1b, racemization of amino acids by AARs proceeds through a direct proton transfer, shuttling between E-PLP and a carbanionic intermediate. In contrast, racemization of amines by AR-OA necessitates supplementation of a cognate ketone to iterate futile cycles between E-PLP and E-PMP, although the formulation of the net chemical equation cancels out the ketone term (e.g., S-**D1** ↔ R-**D1** for $I^S$/$I^R$). Therefore, the cognate ketone acts like a cocatalyst, and the reaction rate should be affected by the ketone level. Indeed, as observed with $II^S/II^R$ and $II^R/II^S$ by ATA-OA in Supplementary Fig. 6b, increasing supplementation of **A1** expedited the racemization of S-**D1** by AR-OA (Supplementary Fig. 15). However, a catalytic amount of the cognate ketone can be generated in the absence of external supply upon mixing only AR-OA and the amine substrate, as oxidative deamination should rapidly proceed until reaching equilibrium of the corresponding elemental reaction. We conjectured that this cosubstrate-free racemization, mimicking the AAR reactions, should be theoretically sound, and indeed, we observed that AR-OA enabled the racemization of S-**D1** without the external supply of **A1** (Supplementary Fig. 16).

## Synthetic applications of AR-OA

To demonstrate the catalytic utility of AR-OA, we carried out the racemization of both enantiomers of 14 chiral amines (10 mM), in the presence of a sub-equimolar cognate ketone (0.5 eq.) and 300 μM AR-OA (Fig. 6a). **D5, D16**, and **D17** underwent racemization most rapidly, leading to less than 5% ee after 1 day. It took 3 days for **D2** and **D14** to reach <10% ee and 7 days for **D1, D3, D6**, and **D18**. **D10-D13** and **D15** showed slower racemization. These results demonstrate the broad substrate specificity of AR-OA, although the ketone and enzyme levels we used were substantially higher than what is ideal for practical applications. However, we anticipate that further engineering of AR-OA to increase $v_{RA}$ for the cognate ketone would enable the use of a reasonable enzyme dosage in conjunction with a catalytic amount of the cognate ketone.

To verify the synthetic potential of AR-OA, we decided to carry out a preparative-scale racemization of 100 mM S-**D5** in the presence of a catalytic amount of a cognate ketone (i.e., 0.1 molar eq.). Prior to the preparative-scale reaction, we examined how the racemization was affected by varying enzyme dosages (30–100 μM) with 100 mM S-**D5** and 10 mM benzylacetone (**A5**), showing gradual increases in the reaction rate as anticipated (Fig. 6b). Considering cost-efficient enzyme use, we opted to use 45 μM AR-OA, corresponding to 4 U/mL, for the preparative-scale reaction. After charging a reaction vessel with 224 mg S-**D5**, 22.2 mg **A5**, and 60 U AR-OA, the reaction mixture was periodically sampled for chiral analysis of **D5** (Fig. 6c). The $ee^S$ value reached 5.6% at 36 h and further decreased to 3.3% after an additional 11-h reaction. The resulting rac-**D5** was subjected to product isolation. The purified rac-**D5**

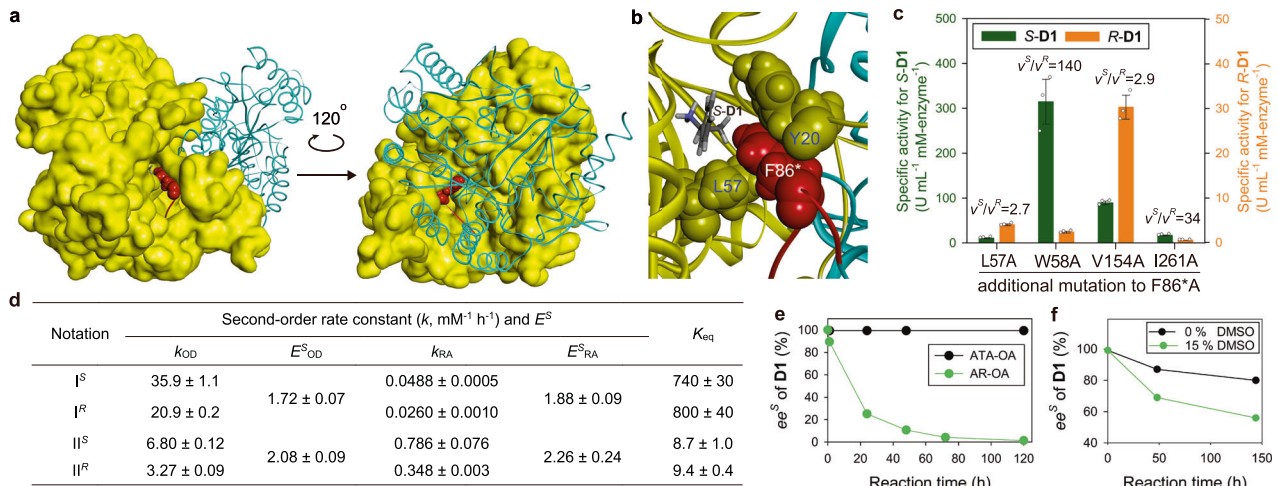

**Fig. 5 | Creation and characterization of AR-OA. a** Homodimeric structure of ATA-OA showing the structural role of F86* in the active-site formation. The subunit comprising most of the active site is shown in surface and the other in ribbon representation. The loop region from F82* to R89* is colored red and F86* is shown in CPK representation. **b** A close-up view of F86* sandwiched between Y20 and L57. **c** Activities of the second-round engineered mutants for *S*-**D1** and *R*-**D1**. Note a ten-fold difference in scale between the left and right y-axis. **d** Stereochemical properties of AR-OA for elemental reactions. **e**, Racemization of *S*-**D1** by AR-OA in comparison with that by ATA-OA. Reaction conditions were 50 mM *S*-**D1**, 50 mM **A1**, and 100 μM enzyme. **f** Effect of DMSO on the racemization of *S*-**D1** by AR-OA. Reaction conditions were 10 mM *S*-**D1**, 5 mM **A1**, 0 or 15 % (v/v) DMSO, and 100 μM AR-OA. Source data are provided as a Source Data file. Data in **c** are mean values of triplicate experiments with error bars indicating the s. d. (*n* = 3).

(53 mg, 23.7% recovery yield, $ee^S$ = 3.4%) was structurally confirmed by $^1$H NMR, $^{13}$C NMR, and LC/MS (Supplementary Fig. 17).

Our results suggest that unleashing the stereochemical trap could activate the dormant AR activity of ATAs for any substrates. As a proof of concept, we tested the racemization of alanine using an ATA-OA mutant that was previously constructed to lose the inherent carboxylate recognition but to improve other native activities through R417A and W58L substitutions[31]. We reasoned that R417A in the L pocket might marginally allow productive binding of D-alanine, whereas the wild type exclusively accepts L-alanine. Indeed, the mutant displayed racemase activity for both enantiomers of alanine (Supplementary Fig. 18).

In summary, this study has uncovered the innate racemase activity of transaminases that is normally under a kinetic trap but can be triggered upon stereochemical relaxation. This might provide insights into the presence of D-amino acids in vivo despite the lack of a specific racemase, e.g. D-aspartate in mammalian brains at nanomolar levels[45]. Our results suggest that transaminases could exhibit racemase activity through iterative futile cycles if enough acceptor is available under certain physiological conditions inducing the stereochemical relaxation. We also demonstrated that stereoselectivity of ATA is amenable to computational engineering and the loss-free racemization of diverse amines can be achieved with a cognate ketone. To the best of our knowledge, this study presents the first example of AR in which the synthetic potential is manifested as the two-base mechanism of natural AARs is not readily applicable to amines[29]. We anticipate that AR-OA or improved variants can offer greater flexibility in the process design for biocatalytic production of chiral amines. Furthermore, our strategy might potentially be extended to a broad range of transaminases beyond ATAs.

## Methods
### Site-directed mutagenesis
Alanine scanning mutants of ATA-OA were previously constructed in a pET28(+) vector[44]. Four mutants, carrying double point mutations, were constructed by a QuikChange Lightning site-directed mutagenesis kit (Agilent Technologies) using the F86*A mutant as a template.

PCR primers used for the mutagenesis were the same as those used for the alanine scanning mutations[44]. The F86*L mutant of ATA-OA and the F85*A mutant of ATA-PD were generated using the same mutagenesis kit with PCR primers designed by the manufacturer's primer design program. Sequences of the mutagenesis primers are provided in Supplementary Table 3. The intended mutagenesis was verified by DNA sequencing.

### Preparation of purified enzymes
For protein expression, we used *Escherichia coli* BL21(DE3) cells transformed with pET28a(+) vectors, harboring ATA-OA[30], ATA-PD[31], ATA-CV[31], ATA-AR[46], and ATA-AF[46] genes, as described elsewhere. Cell cultivation and purification of the His-tagged ATAs were performed as described previously[31]. Detailed experimental procedures are presented in the Supplementary Methods. Molar concentrations of the purified ATAs were determined by UV absorbance at 280 nm using molar extinction coefficients obtained at http://www.biomol.net/en/tools/proteinextinction.htm.

### Enzyme activity assay
Unless otherwise specified, all the enzyme assays to measure ATA activity were carried out at 37 °C and pH 7 (50 mM potassium phosphate buffer) under initial rate measurement conditions (i.e., conversion <20%). One unit of ATA is defined as the enzyme amount required to produce 1 μmol of **A1** in 1 min at 10 mM *S*-**D1** and 10 mM pyruvate. **A1** produced was analyzed by HPLC. Microsoft Excel was used for analyzing enzyme activities.

### Stereoinversion of *S*-D2 by ATA-OA in the presence of a non-cognate cosubstrate
Unless otherwise specified, all reactions were carried out with 50 mM amino donor, 50 mM acceptor, and 750 μM ATA-OA at 37 °C and pH 7 (50 mM phosphate buffer supplemented with 0.5 mM PLP). Regarding the reactions between **A2** and **D3**-**D9**, 650 μM enzyme was used and the reaction mixture contained 15% (v/v) DMSO. Racemic forms of **D3**-**D9**, except for **D8** and **D9**, were used as the amino donor. At predetermined reaction times, the reaction samples were taken for chiral HPLC analysis of **D2**.

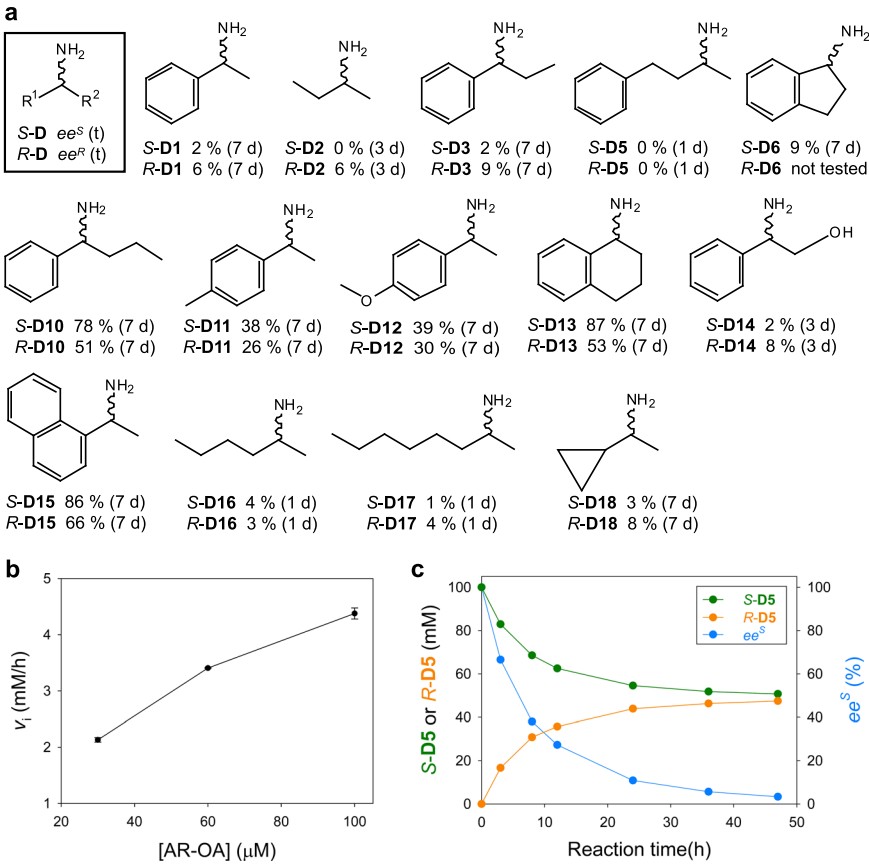

**Fig. 6 | Synthetic utility of AR-OA. a** Racemization of diverse chiral amines using AR-OA in the presence of a cognate ketone. **b** Effect of the enzyme concentration on the initial formation rate of $R$-**D5** from $S$-**D5**. **c** Preparative-scale conversion of $S$- **D5** to *rac*-**D5**. Source data are provided as a Source Data file. Data in **b** are mean values of triplicate experiments with error bars indicating the s. d. ($n = 3$).

## Kinetic analysis

To determine second-order rate constants, a pseudo-one-substrate kinetic model used in a previous study[30] was further simplified under limiting conditions (see Supplementary Methods for details). Briefly, reaction conditions were set to meet the limiting conditions where the concentrations of substrate and cosubstrate were lower and higher, respectively, than the corresponding $K_M$ values. The resulting equation, i.e. $k \approx v_i/([E][S])$, was used to determine $k$ with three independent initial rate data ($v_i$) measured at fixed concentrations of enzyme (E) and substrate (S). Linear regression of the product formation against time was used to determine the $v_i$ values. The cosubstrates used to determine $k_{OD}$ and $k_{RA}$ were pyruvate and **D8**, respectively. Specific reaction conditions and products analyzed for the kinetic measurements are listed in Supplementary Table 4. Initial rates for the kinetic analysis were determined at <20% conversion. Kinetic analysis and preparation of charts were carried out using Sigmaplot.

## Spectral analysis

The E-PLP form of ATA-OA was prepared by incubation of the purified enzyme with 10 mM pyruvate for 1 h at 24 °C and then desalting on a HiTrap column (GE healthcare) using a 20 mM sodium phosphate buffer (pH 7) supplemented with 75 mM NaCl. The resulting E-PLP (11 µM) was mixed with each enantiomer of **D2** (50 mM), followed by incubation at 24 °C until the aliquot was sampled for spectral measurement under a wavelength scanning mode of a UV-1650PC spectrophotometer (Shimadzu) using UVProbe 2.62. The scanning performed immediately after the mixing was designated by the 0.1-min incubation.

## Mathematical modeling of enzyme reactions

Numerical simulations of the ATA-OA reaction require mass balance equations for related chemical species and enzyme intermediates whose time derivatives can be expressed using a second-order kinetics as described previously[47]. Briefly, simulations of the reaction between $S$-**D1** and **A2** involve $I^S$, $II^S$, and $II^R$, and thus should include balance equations for $S$-**D1**, **A1**, $S$-**D2**, $R$-**D2**, **A2**, E-PLP, and E-PMP. The resulting seven differential equations were numerically solved using Mathematica 12 (Wolfram Research). Regarding the AR reaction between **D2** and **A2**, only $II^S$ and $II^R$ are included in the simulation using time derivatives of $S$-**D2**, $R$-**D2**, **A2**, E-PLP, and E-PMP. Detailed experimental procedures are presented in the Supplementary Methods.

## Direct racemization of D2 with A2 by native ATAs

Unless otherwise specified, all reactions were carried out with 50 mM $S$-**D2** or $R$-**D2**, 50 mM **A2**, and 100 µM ATA at 37 °C and pH 7 (50 mM phosphate buffer containing 0.5 mM PLP). The reactions at varying concentrations of **A2** were carried out at 10 mM $S$-**D2** or $R$-**D2**, 1- 20 mM **A2**, and 50 µM ATA-OA. Regarding ATAs other than ATA-OA, we used 100 µM enzyme for the reactions.

## Molecular modeling

Molecular modeling was performed with the Discovery Studio package (version 4.5, Accelrys). Docking simulations require an E-PLP structure harboring the active-site arginine in an outward conformation. However, the crystal structure of ATA-OA available (PDB ID: 5GHF) is an E-PMP form assuming an inward R417[31]. Therefore, we conducted structural modifications of 5GHF to construct the desired structure by recruiting the PLP and the arginine structures from ATA-CV and

ATA-PD, respectively, as described elsewhere[44]. The resulting structure was further modified to a tautomeric structure of the internal aldimine where the Nε and the phenolic oxygen were protonated and deprotonated, respectively, as shown in Fig. 1b. Protonated forms of **D1** and **D2** were used as ligands for docking simulations on the CDOCKER module under a default setting (2,000 steps at 700 K for a heating step; 5000 steps at 300 K for a cooling step; 8 Å grid extension) within the active site defined by the Binding Site module. The alanine scanning mutants of ATA-OA were generated by the Design Protein module of the Discovery Studio package without additional energy minimization.

### Racemization of various chiral amines using AR-OA
All reactions were carried out with 10 mM *S*- or *R*-amine, 5 mM cognate ketone, and 300 µM AR-OA at 37 °C and pH 7 (50 mM Tris buffer containing 0.5 mM PLP). DMSO (15% (v/v)) was included in the reactions with *S*-amines. *R*-**D6** could not be tested due to the unavailability of the chemical stock. Reaction mixtures were incubated in reaction vials which were kept tightly sealed to avoid evaporation of ketone. Aliquots of the reaction mixture were sampled after 1, 3, and 7 days, and then subjected to chiral HPLC analysis of the amine substrates.

### Preparative-scale racemization of *S*-D5 and product isolation
Reaction conditions were 100 mM *S*-**D5**, 10 mM **A5**, and 45 µM AR-OA in 50 mM potassium phosphate (pH 7) supplemented with 0.5 mM PLP (total reaction volume = 15 mL) at 37 °C under magnetic stirring. For product isolation, the pH of the reaction mixture was adjusted to 1.9 after a 47-h reaction by adding 5 N HCl solution (110 µL), and the protein precipitate formed was removed by centrifugation (13,000 × g, 40 min). The resulting supernatant was treated with two extractions of 40 mL hexane each to remove **A5**. The pH of the ketone-free reaction mixture was adjusted to 12 by adding 1.4 mL of 5 N NaOH solution, and the deprotonated **D5** was extracted three times, each with 40 mL of hexane. The extractant pool was vacuum evaporated at 50 °C and 450 hPa, resulting in liquid **D5**. The isolated **D5** was characterized by $^1$H NMR, $^{13}$C NMR, and LC/MS. $^1$H and $^{13}$C NMR spectra were recorded on an Avance III HD 400 spectrometer (Bruker Co.). Mass spectral data were obtained with a Thermo Q-Exactive Orbitrap mass spectrometer connected to a Dionex Ultimate 3000 Nano HPLC (Thermo Fisher Scientific Inc.).

### HPLC analysis
HPLC analysis was performed using an Alliance system (Waters) or a 1260 Infinity II LC System (Agilent Technologies). **A1** was analyzed on a Symmetry C18 column (Waters) using isocratic elution of 60/40/0.1% (v/v) MeOH/water/trifluoroacetic acid at a flow rate of 1 mL/min under UV detection tuned at 254 nm. Chiral analysis of alkylamines and alanine was carried out after derivatization with a Marfey's reagent as described previously[46]. Detailed experimental procedures are presented in the Supplementary Methods. A Crownpak CR(-) column (Daicel) was used for chiral analysis of arylalkylamines. HPLC data were collected by EMPOWER Pro and OpenLAB for Waters and Agilent HPLC, respectively. Detailed chiral analysis conditions are provided in Supplementary Table 5.

### Other related methods
ChemDraw and Adobe illustrator were used to draw chemical structures and to combine figures, respectively. Detailed descriptions of chemicals information, kinetic analysis, cell cultivation and enzyme purification, mathematical modeling, and chiral analysis using a Marfey's reagent are provided as Supplementary Methods in the Supplementary Information.

### Reporting summary
Further information on research design is available in the Nature Portfolio Reporting Summary linked to this article.

### Data availability
The data supporting the findings of this study are included in the article and the Supplementary Information. The structural data of ATA-OA used in this study is from Protein Data Bank under accession code 5GHF. All unique biological materials (plasmids and strains) are readily available from the corresponding authors upon request. Source data are provided with this paper and have been deposited in figshare under https://doi.org/10.6084/m9.figshare.24217176. Source data are provided with this paper.

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

## Acknowledgements

This work was supported by the National Research Foundation of Korea (NRF) funded by the Korea government (MSIT) (Grant No. 2021R1A2C2013069). S.-W. Han was financially supported by Initiative for Biological Function & Systems under the BK21 PLUS program of Korean Ministry of Education. We thank Jimin Song for assistance during molecular modeling.

## Author contributions

J.-S.S. conceived the idea, supervised the project, and wrote the manuscript. S.-W.H. developed the idea and undertook most experiments and data analysis. Y.J. carried out the ketone-free racemization of *S*-**D1**, wavelength scanning of AR-OA, and racemization of alanine. J.K. carried out the activity measurements of ATA-OA mutants carrying double point mutations. J.J. performed preparation of the ATA-OA/F86*L mutant.

## Competing interests

The authors declare no competing interests.
