## [Peer Review File · Nature Communications]

REVIEWER COMMENTS

Reviewer #1 (Remarks to the Author):

Racemization of sturdy amines (i.e., unfunctionalized) is more challenging than that of alcohols, and its importance is well described by Musa M Musa (Chirality 2020, 32, 147-157). So, it is highly desirable to create amine racemase, which has yet to be discovered in nature, in the amine chemistry and biotransformation field. Faber et al. reported a biocatalytic protocol and its implementation in the racemization of sturdy enantiopure α -chiral primary amines. They used a pair of stereo-complementary transaminases (TA) in this approach, which relies on the interconversion of the enantiomers of an amine and their corresponding ketone via reversible deamination/amination reactions (Chem A Eur J. 2011; 17: 378-383). This strategy was recently further improved using the broadly accepted alanine/pyruvate cosubstrate system. (ChemCatChem. 2018; 10(21): 5012- 5018). However, both cases have utilized two TAs ((R)- and (S)-selective), which give complexity, difficulty in controlling the reaction, and limited applicability. The ideal way is to generate an amine racemase. In this study, engineered TA showed good racemase activity toward both enantiomers of diverse chiral amines in the presence of a cognate ketone. This strategy may generally apply to a wide range of TAs, paving the way for developing new-to-nature racemases. The experiments are designed very well, and the manuscript is written nicely. It is acceptable after revising it according to the comments.

Comments

1. AR-OA enables racemization of S-D1 under cosubstrate-free conditions (Fig S13). In “Synthetic applications of AR-OA,” racemization was performed in the presence of 0.5 eq cognate ketone; it is a lot. Non-use of ketone or the catalytic amount of ketone is ideal. The authors need to show the experimental results in the absence or the presence of the catalytic amount of ketone. Moreover, based on that, it should be concluded why extra ketone is needed in the reaction.
2. In the “Kinetic dissection of ATA-OA.” The authors conducted a kinetic analysis of ATA-OA with each enantiomer of D1 and D2 to investigate the stereochemical properties. For example, S-D1 + A2 \leftrightarrow A1 + S-D2 is denoted by IS /IIS, which represents IS coupled with the reverse of IIS. The approach is quite nice and impressive.
3. In Fig 1 B, in the “deprotonation” and “protonation” steps, the proton (H⁺, not H) should be presented.
4. For the racemization of various chiral amines using AR-OA, the authors used 300 μ M of the enzyme, and the reaction time was seven days; it was quite a lot. It should be discussed in the manuscript. However, using a lot of enzyme will not dilute the scientific impact of current work. With a similar line, the author utilized ATA from *Ochrobactrum anthropi* as a parent enzyme to generate racemase. The selection of parent enzyme is crucial to obtain desirable mutant. What will happen if you introduce the same mutations into other TAs? Thus, the transposability of these mutations to other (S)-ATAs could support the experimental data in this study.

5. To accept R-D1, docking simulations were performed on alanine mutants, and F86 was identified as a promising residue. Recently, the Bornscheuer group reported engineering of ATA from *Ruegeria* sp. (3FCR), and variants reported herein also showed both (R-) and (S-) stereo references toward a broad range of substrates (<https://doi.org/10.1002/ange.202301660>). Mutations to F86 to other residues like Leu should also be tested for enhanced AR-OA activity.

Reviewer #2 (Remarks to the Author):

The manuscript has a very interesting approach on repurposing an amine transaminase into an amine racemase. The study is very interesting from a mechanistic point of view, but as well from an evolutionary point of view, as amine racemases are not common in nature.

The authors did a very thorough analysis of the mechanism and the amino acids that are involved in the stages required for the repurposing of the promiscuous amine transaminase of interest. The results are solid and indeed they could repurpose the transaminase to racemase and reverse also the effect later.

There are a few points that need clarification in order to maximize the impact of this work:

- 1) In the assays for the racemization ("racemization of various chiral amines using AR-OA") it is mentioned that the cognate ketone is added in the reaction. A typical racemase does not require the ketone. Can the authors elaborate on that, why is the ketone required in the reaction?
- 2) Mutagenesis on homologous enzymes of variable percentage of identity would be interesting, to show if the hypothesis has a wider applicability. As the authors state that "Furthermore, our strategy might potentially be extended to a broad range of transaminases beyond ATAs." I fail to see already the applicability in ATAs, I cannot see if this is applicable in other transaminases. Are the positions targeted in this work conserved in families or subfamilies of amine transaminases or transaminases in general? How would the authors expand the scope of the work?
- 3) In the same direction, did the authors try to perform reverse engineering according to their hypothesis, to convert an amine racemase to an amine transaminase?
- 4) The scale on the synthetic applications is quite low. How come and the authors did not intensify the process, in order to highlight the synthetic potential? Is there any inhibition observed? This scale is not industrially relevant, if we only consider this one enzyme that was evolved. Isolated yields are required, as well as full characterization of the final products.

So, in general, I believe that the authors should focus either in proving their wider applicability of the hypothesis and that this rational approach works in other enzymes too, or to strengthen the synthetic part, in order to highlight the synthetic potential of the engineered enzyme.

Reviewer #3 (Remarks to the Author):

This manuscript reports the discovery of native amine racemization activity within amine transaminases (ATAs). In the presence of ketone co-substrates, both S- and R-configured amines have been demonstrated to undergo slow racemization resulting in near racemic mixtures after periods of time ranging from 1-7 days. Computational and structure guided active-site engineering has led to the identification of some key residues implicated in the racemization activity and also allowed the authors to build a kinetic model.

There is no doubt that the development of amine racemase enzymes is of great interest for preparative biocatalysis. As the authors point out, in principle an amine racemase could be combined with an enantioselective lipase to effect a dynamic kinetic resolution (DKR) of racemic amines which would have considerable practical value since currently such processes require external racemization of the unreacted amine.

However lipase catalyzed acylation reactions are typically carried out in low water activity environments (e.g. organic solvents) and it is not clear if the amine racemases developed in this work would be active under these conditions. This is especially the case since for many of the substrates the activities are already quite low and will probably be lower still under low water activity conditions.

Nevertheless I would like to see the authors propose and carry out some initial studies in which they combine their alanine racemases with a second enzyme system to achieve a DKR process, however inefficient at this stage. Another issue could well be the require for high concentrations of ketone which may complicate the overall process.

Some corrections that need to be made:

superscripts throughout for R1, R2 etc.

Response to the reviewer's comments

We appreciate the reviewer's comments for the improvement of the manuscript. We have prepared a revision to accommodate the reviewer's comments. All the changes in the revision are marked in green shade.

First, we demonstrated the generality, in terms of transposability and variability, of the F86* residue as a mutation spot to elicit stereochemical promiscuity in *S*-selective ATAs. We conducted a multiple sequence alignment of 14 *S*-selective ATAs and prepared an F85*A mutant of ATA-PD and an F86*L mutant of ATA-OA, which were assayed for *S*-**D1** and *R*-**D1**. This led to the presentation of new results in the revision as Supplementary Table 2 and Supplementary Figures 11 and 12.

Second, to clarify the necessity of excess ketone, the discussion on Supplementary Figure 13 in the previous manuscript (moved to Supplementary Figure 16 in the revision) has been completely revised. For better explanation, we conducted additional experiments to show that the racemization reaction by AR-OA is expedited by increasing the supplementation of a cognate ketone (Supplementary Figure 15 in the revision).

Third, to demonstrate the scalability of the racemization reaction, we carried out preparative-scale racemization of *S*-**D5** (Figure 6c in the revision) and then isolated and characterized the resulting *rac*-**D5** (Supplementary Figure 17 in the revision). In addition, we conducted an additional experiment to examine the effect of enzyme concentration on the racemization of *S*-**D5** (Fig. 6b in the revision) prior to the preparative scale reaction. To demonstrate the use of a catalytic amount of the cognate ketone, the preparative-scale racemization of *S*-**D5** (100 mM) was conducted in the presence of 0.1 molar eq. of benzylacetone (i.e., 10 mM).

We believe that the revised manuscript, including the additional results, meets all the reviewer's comments and suggestions.

< Reviewer 1 >

1. Comment Racemization of sturdy amines (i.e., unfunctionalized) is more challenging than that of alcohols, and its importance is well described by Musa M Musa (Chirality 2020, 32, 147-157). So, it is highly desirable to create amine racemase, which has yet to be discovered in nature, in the amine chemistry and biotransformation field. Faber et al. reported a biocatalytic protocol and its implementation in the racemization of sturdy enantiopure α -chiral primary amines. They used a pair of stereo-complementary transaminases (TA) in this approach, which relies on the interconversion of the enantiomers of an amine and their corresponding ketone via reversible deamination/amination reactions (Chem A Eur J. 2011; 17: 378-383). This strategy was recently further improved using the broadly accepted alanine/pyruvate cosubstrate system. (ChemCatChem. 2018; 10(21): 5012-5018). However, both cases have utilized two TAs ((*R*)- and (*S*)-selective), which give complexity, difficulty in controlling the reaction, and limited applicability. The ideal way

is to generate an amine racemase. In this study, engineered TA showed good racemase activity toward both enantiomers of diverse chiral amines in the presence of a cognate ketone. This strategy may generally apply to a wide range of TAs, paving the way for developing new-to-nature racemases. The experiments are designed very well, and the manuscript is written nicely. It is acceptable after revising it according to the comments.

Response We appreciate the reviewer's comments. We added the paper by Musa as reference 23.

2. Comment AR-OA enables racemization of *S*-**D1** under cosubstrate-free conditions (Fig S13). In "Synthetic applications of AR-OA," racemization was performed in the presence of 0.5 eq cognate ketone; it is a lot. Non-use of ketone or the catalytic amount of ketone is ideal. The authors need to show the experimental results in the absence or the presence of the catalytic amount of ketone. Moreover, based on that, it should be concluded why extra ketone is needed in the reaction.

Response We appreciate the reviewer's comments. We realized that the explanation of Fig. S13 in the previous manuscript, now moved to Fig. S16 in the revision, could be misleading. Since the cognate ketone is an essential component in the two elemental reactions comprising the complete reaction for racemization of amine, iterative futile cycles between E-PLP and E-PMP are not possible in the absence of the cognate ketone, and thus racemization cannot proceed. Therefore, as already shown in Fig. 3e and Fig. S6b with the wild-type ATA-OA, increases in the ketone level expedite the racemization reaction. To corroborate this, we performed an additional experiment to examine the improvement of the racemization of *S*-**D1** by AR-OA in response to increasing **A1** supplementation (Fig. S15 in the revision).

What we meant by Fig. S13 in the previous manuscript was that, even in the absence of external supply, the cognate ketone can be generated upon mixing only AR-OA and amine because of spontaneous oxidative deamination. For example, as shown in Fig. 5d, the equilibrium constants for I^S and I^R are higher than 700, and consequently, most of the E-PLP form of AR-OA should be converted to E-PMP upon being mixed with **D1**. This leads to a rapid generation of **A1**, nearly up to the level of the initial amount of E-PLP in the enzyme dosage. This is why racemization of *S*-**D1** could proceed without the external supply of **A1**, as shown in Fig. S13 in the previous manuscript. Under the reaction conditions without the external ketone supply, the amount of ketone generated during the initial oxidative deamination should be lower than the enzyme dosage. This is why we used a high AR-OA concentration, specifically 1.44 mM, to achieve a satisfactory racemization rate.

Concerning the use of a catalytic amount of the cognate ketone, we conducted a preparative-scale racemization of *S*-**D5** (100 mM) in the presence of 10 mM benzylacetone (0.1 molar eq.) and 45 μ M AR-OA (Fig. 6c in the revision). Racemization was completed within 2 days and the resulting *rac*-**D5** ($ee^S = 3.3\%$) was purified and structurally characterized (Fig. S17 in the revision).

Taken all together, this part has been completely revised in lines 357-370 and 382-393.

3. Comment In the “Kinetic dissection of ATA-OA.” The authors conducted a kinetic analysis of ATA-OA with each enantiomer of D1 and D2 to investigate the stereochemical properties. For example, $S\text{-D1} + A2 \leftrightarrow A1 + S\text{-D2}$ is denoted by IS /IIS, which represents IS coupled with the reverse of IIS. The approach is quite nice and impressive.

Response We appreciate the reviewer’s comments.

4. Comment In Fig 1 B, in the “deprotonation” and “protonation” steps, the proton (H^+ , not H) should be presented.

Response We appreciate the reviewer’s comment. Fig. 1b has been corrected as the reviewer kindly pointed out.

5. Comment For the racemization of various chiral amines using AR-OA, the authors used 300 μ M of the enzyme, and the reaction time was seven days; it was quite a lot. It should be discussed in the manuscript. However, using a lot of enzyme will not dilute the scientific impact of current work. With a similar line, the author utilized ATA from *Ochrobactrum anthropi* as a parent enzyme to generate racemase. The selection of parent enzyme is crucial to obtain desirable mutant. What will happen if you introduce the same mutations into other TAs? Thus, the transposability of these mutations to other (S)-ATAs could support the experimental data in this study.

Response We appreciate the reviewer’s comments. As the reviewer pointed out, we agree that the 300 μ M enzyme used in Fig. 6 in the previous manuscript (now moved to Fig. 6a in the revision) is high. In addition to the high ketone supplementation, we have addressed these problems and a possible solution in lines 377-381 in the revision. Furthermore, this reviewer’s comment led us to examine the effect of AR-OA concentration on the racemization of *S*-**D5** (Fig. 6b in the revision), prior to the preparative scale reaction (Fig. 6c in the revision). As expected, the higher the enzyme level is, the faster the

racemization proceeds. As a compromise for reasonable enzyme usage, we employed 45 μ M AR-OA for the preparative scale reaction.

Our study indicates that a stereochemically promiscuous transaminase can act as a racemase for native substrates. Our results with ATA-OA suggest that the gatekeeper residue against stereochemical promiscuity is F86*. To examine the transposability of the F86*A mutation to allow for stereochemical promiscuity, we conducted a partial multiple sequence alignment of 14 *S*-selective ATAs whose X-ray structures have been determined (Table S2 in the revision). Indeed, the F86* residue of ATA-OA is conserved and represented by aromatic amino acids, specifically eleven Phe, two Tyr, and one Trp. We chose ATA-PD to verify the transposability of the F86*A mutation and introduced the alanine substitution at the same spot. The resulting F85*A mutant of ATA-PD showed a drastic reduction in the v^S/v^R value for **D1**, decreasing from 4100 to 270 (Fig. S11 in the revision). In addition, an alanine substitution at the same spot in ATA-CV was reported elsewhere to display a change in the *E* value from 150 to 7 (added to the revision as reference 42). Moreover, according to a recent paper that the reviewer indicated, a point mutation to leucine at the same position resulted in ATA from *Ruegeria* sp. TM1040 exhibiting substantial activity for *R*-**D1** (added to the revision as reference 43). Taken all together, these results suggest that mutation of the residue corresponding to F86* of ATA-OA is generally applicable to *S*-selective ATAs. These points have been mentioned in lines 304-323.

6. Comment To accept R-D1, docking simulations were performed on alanine mutants, and F86 was identified as a promising residue. Recently, the Bornscheuer group reported engineering of ATA from *Ruegeria* sp. (3FCR), and variants reported herein also showed both (R-) and (S-) stereo references toward a broad range of substrates (<https://doi.org/10.1002/ange.202301660>). Mutations to F86 to other residues like Leu should also be tested for enhanced AR-OA activity.

Response We appreciate this comment. The motivation for the second round engineering to create AR-OA was based on the structural speculation that the F86*A might induce destabilization of the neighboring residues due to creation of a cavity. In addition to the reviewer's suggestion, we realize that F86*L is an ideal mutation to test whether a milder structural perturbation could lead to a less activity loss for *S*-**D1**. We prepared the F86*L mutant of ATA-OA and measured the activity for *S*-**D1** and *R*-**D1** (Fig. S12 in the revision). Indeed, the F86*L mutant showed only a 20 % reduction in native activity for *S*-**D1**. In agreement with the Y87*L mutant of ATA from *Ruegeria* sp. TM1040, the

F86*L mutant of ATA-OA allows for substantial activity for *R-D1*, resulting in $v^S/v^R = 91$. This part has been added to lines 331-336 in the revision.

< Reviewer 2 >

1. Comment The manuscript has a very interesting approach on repurposing an amine transaminase into an amine racemase. The study is very interesting from a mechanistic point of view, but as well from an evolutionary point of view, as amine racemases are not common in nature. The authors did a very thorough analysis of the mechanism and the amino acids that are involved in the stages required for the repurposing of the promiscuous amine transaminase of interest. The results are solid and indeed they could repurpose the transaminase to racemase and reverse also the effect later.

Response We appreciate the reviewer's comments.

2. Comment There are a few points that need clarification in order to maximize the impact of this work: 1) In the assays for the racemization ("racemization of various chiral amines using AROA") it is mentioned that the cognate ketone is added in the reaction. A typical racemase does not require the ketone. Can the authors elaborate on that, why is the ketone required in the reaction?

Response We appreciate the reviewer's comments. We realized that the previous manuscript did not provide sufficient explanation on why ketone is needed for the racemization reaction. To clarify this point, we completely revised the discussion of Fig. S13 in the previous manuscript (now moved to Fig. S16 in the revision). In brief, unlike the amino acid racemase, the amine racemization requires the cognate ketone as a cocatalyst and thereby increasing the ketone supplementation expedites the reaction rate. Detailed discussion has been added to lines 357-370.

3. Comment 2) Mutagenesis on homologous enzymes of variable percentage of identity would be interesting, to show if the hypothesis has a wider applicability. As the authors state that "Furthermore, our strategy might potentially be extended to a broad range of transaminases beyond ATAs." I fail to see already the applicability in ATAs, I cannot see if this is applicable in other transaminases. Is the positions targeted in this work conserved in families or subfamilies of amine transaminases or transaminases in general? How would the authors expand the scope of the work?

Response We appreciate the comments. Our results with ATA-OA suggest that the F86* residue serves as a gatekeeper against stereochemical promiscuity. As the reviewer recommended, we carried out a partial multiple sequence alignment of 14 *S*-selective ATAs whose X-ray structures have been determined (Table S2 in the revision). Indeed, the F86* residue of ATA-OA is conserved and represented by aromatic amino acids, specifically eleven Phe, two Tyr, and one Trp. We chose ATA-PD to verify the transposability of the F86*A mutation and introduced the alanine substitution at the same spot. The resulting F85*A mutant of ATA-PD showed a drastic reduction in the v^S/v^R value from 4100 to 270 (Fig. S11 in the revision). In addition, the alanine substitution at the same spot of ATA-CV was reported elsewhere to display a change in the *E* value from 150 to 7 (reference 42 in the revision). Ao et al. recently reported that a point mutation at the same position to leucine resulted in ATA from *Ruegeria* sp. TM1040, also included in Table S2, exhibiting substantial activity for *R-D1* (reference 43 in the revision). Taken all together, these results suggest that mutation of the residue corresponding to F86* of ATA-OA can be generally applicable to *S*-selective ATAs. These points have been mentioned in lines 304-323.

4. Comment 3) in the same direction, did the authors try to perform reverse engineering according to their hypothesis, to convert an amine racemase to an amine transaminase?

Response We appreciate the reviewer's comments. What we understand from this comment is whether we carried out engineering of an amino acid racemase (AAR) to an amine transaminase (ATA). Although this is an interesting topic, we have not attempted AAR-to-ATA engineering, as explained the "Theoretical feasibility of AR by engineering natural enzymes" section.

AR-OA is basically an amine transaminase whose stereospecificity has been engineered to be promiscuous. AR-OA retains its transaminase activity, and the stereochemically promiscuous futile cycles between E-PLP and E-PMP lead to the racemization of amines.

5. Comment 4) The scale on the synthetic applications is quite low. How come and the authors did not intensify the process, in order to highlight the synthetic potential? Is there any inhibition observed? This scale is not industrially relevant, if we only consider this one enzyme that was evolved. Isolated yields are required, as well as full characterization of the final products.

Response We appreciate the reviewer's comments. In relation to the scale of the synthetic applications, we agree with the reviewer's concerns. To address this issue, we conducted a preparative-scale racemization of *S-D5* (100 mM) in the presence of 10 mM

benzylacetone (0.1 molar eq.) and 45 μ M AR-OA (Fig. 6c in the revision). The racemization was completed within 2 days, and the resulting *rac*-**D5** was subsequently purified and structurally characterized using ^1H NMR, ^{13}C NMR, and LC/MS (Fig. S17 in the revision). This part has been mentioned in lines 382-393 in the revision.

With respect to inhibition, the racemization of **D2** by ATA-OA did not show any substrate inhibition as already shown in Fig. 3e and Supplementary Fig. 6. We also conducted an additional experiment to examine how racemization of *S*-**D1** (100 mM) by AR-OA is affected by increasing the **A1** supplementation (5-50 mM). The racemization reaction rate showed a positive dependency on the ketone concentration (Fig. S15 in the revision), corroborating the absence of substrate inhibition.

6. Comment So, in general, I believe that the authors should focus either in proving their wider applicability of the hypothesis and that this rational approach works in other enzymes too, or to strengthen the synthetic part, in order to highlight the synthetic potential of the engineered enzyme.

Response We appreciate the reviewer's comments. As stated in our responses to comments 3 and 5, we have demonstrated the transposability of the F86*A mutation to other ATAs using the F85*A mutant of ATA-PD and have also shown the scalability of the AR-OA reaction through the preparative-scale racemization of *S*-**D5**. We believe that these additional results adequately address the reviewer's comments.

< Reviewer 3 >

1. Comment This manuscript reports the discovery of native amine racemization activity within amine transaminases (ATAs). In the presence of ketone co-substrates, both *S*- and *R*-configured amines have been demonstrated to undergo slow racemization resulting in near racemic mixtures after periods of time ranging from 1-7 days. Computational and structure guided active-site engineering has led to the identification of some key residues implicated in the racemization activity and also allowed the authors to build a kinetic model.

Response We appreciate the reviewer's comments.

2. Comment There is no doubt that the development of amine racemase enzymes is of great interest for preparative biocatalysis. As the authors point out, in principle an amine racemase could be combined with an enantioselective lipase to effect a dynamic kinetic resolution (DKR) of racemic amines which would have considerable practical value since currently such processes require external racemization of the unreacted amine.

However lipase catalyzed acylation reactions are typically carried out in low water activity environments (e.g. organic solvents) and it is not clear if the amine racemases developed in this work would be active under these conditions. This is especially the case since for many of the substrates the activities are already quite low and will probably be lower still under low water activity conditions.

Nevertheless I would like to see the authors propose and carry out some initial studies in which they combine their alanine racemases with a second enzyme system to achieve a DKR process, however inefficient at this stage.

Response We appreciate the reviewer's comments. In this study, our primary focus was on understanding the racemization of **D2** by native amine transaminases and on expanding this finding to create an amine racemase by imparting stereochemical promiscuity to a native transaminase. The demonstration of the DKR process using AR-OA is a topic we plan to explore for a future research. We believe that development of the DKR process should be technically feasible as the enzyme cascade of lipase and ATA in organic solvent has been demonstrated elsewhere for chiral amine synthesis from a ketone precursor (Green Chem., 2023, 25, 6041-6050; Adv. Synth. Catal., 2013, 355, 1703–1708). We hope the reviewer understands our decision in this regard.

3. Comment Another issue could well be the require for high concentrations of ketone which may complicate the overall process.

Response We appreciate the reviewer's comments. The cognate ketone is an essential component in the two elemental reactions comprising the complete reaction for racemization of amine. Therefore, as already shown in Fig. 3e and Fig. S6b with the wild-type ATA-OA, increases in the ketone level expedite the racemization reaction. To corroborate this, we performed an additional experiment to examine the improvement of the racemization of *S*-**D1** by AR-OA in response to increasing **A1** supplementation (Fig. S15 in the revision). However, as the reviewer pointed out, high concentrations of ketone complicate the overall process. To prove use of a catalytic amount of the cognate ketone, we conducted a preparative-scale racemization of *S*-**D5** (100 mM) in the presence of 0.1 molar eq. of benzylacetone (Fig. 6c in the revision). Racemization was completed within 2 days and the resulting *rac*-**D5** was purified and structurally characterized (Fig. S17 in the revision). These parts have been added to lines 357-370 and 382-393 in the revision.

4. Comment Some corrections that need to be made: superscripts throughout for R1, R2 etc.

Response We appreciate the reviewer's comments. We corrected Fig. 1a and Fig. 6a as the reviewer recommended.

REVIEWERS' COMMENTS

Reviewer #1 (Remarks to the Author):

This reviewer appreciates the authors' efforts to revise it and address the suggested comments.

All comments have been well addressed, and now it is acceptable for publication.

Reviewer #2 (Remarks to the Author):

Taking into considerations the revised manuscript, as well as the rebuttal letter of the authors, I believe that the authors addressed all topics mentioned; especially (1) the role of the ketone in the reaction, while they even performed semi-preparative experiments with catalytic amount of the ketone, and (2) proving the applicability of the mutation suggested to other enzymes. These additions / clarifications significantly improved the quality and clarity of the manuscript, which I now believe is in a state to be published, as is.

Reviewer #3 (Remarks to the Author):

I am satisfied that the authors have addressed most, if not all, of the comments made by the referees, especially the effect of ketone concentration on the rate of racemization. On this basis I am happy to recommend publication.